# The National Implementation of a Community Pharmacy Antimicrobial Stewardship Intervention (PAMSI) through the English Pharmacy Quality Scheme 2020 to 2022

**DOI:** 10.3390/antibiotics12040793

**Published:** 2023-04-21

**Authors:** Catherine V. Hayes, Sejal Parekh, Donna M. Lecky, Jill Loader, Carry Triggs-Hodge, Diane Ashiru-Oredope

**Affiliations:** 1HCAI, Fungal, AMR, AMU & Sepsis Division, UK Health Security Agency, London SW1P 3HX, UK; 2Primary Care Strategy and NHS Contracts Group, Primary, Community and Personalised Care Directorate, NHS England, London SE1 8UG, UK

**Keywords:** community pharmacist, questionnaire, antimicrobial resistance, behavioural, incentive, e-Learning, antimicrobials, infections, primary care, community healthcare, pharmacy technicians

## Abstract

Since 2020, England’s Pharmacy Quality Scheme (PQS) has incentivised increased antimicrobial stewardship (AMS) activities in community pharmacy. In 2020/21, this included the requirement for staff to complete an AMS e-Learning module, pledge to be an Antibiotic Guardian and develop an AMS Action plan. To build and embed these initiatives, in 2021/22, the PQS required the use of the TARGET Antibiotic Checklist (an AMS tool for use when patients present with a prescription for antibiotics to support conducting and recording of a series of safety and appropriateness checks against each prescribed antibiotic). This paper describes the implementation of the national PQS criteria from 2020 to 2022, and details community pharmacies’ AMS activities and barriers to implementation of the 2021/22 criteria. A total of 8374 community pharmacies submitted data collected using the TARGET Antibiotic Checklist for 213,105 prescriptions; 44% surpassed the required number for the PQS. Pharmacy teams reported checking the following: duration, dose, and appropriateness of antibiotics; patient allergies and medicine interactions (94–95%); antibiotic prescribing guideline adherence (89%); and the patient’s previous use of antibiotics (81%). The prescriber was contacted for 1.3% of TARGET Antibiotic Checklists (2741), and the most common reasons for such contacts were related to dose, duration, and possible patient allergy. A total of 105 pharmacy staff responded to a follow-up questionnaire, which suggested that some AMS principles had been embedded into daily practice; however, the necessary time commitment was a barrier. The PQS was able to incentivise mass AMS activities at pace over consecutive years for England’s community pharmacies simultaneously. Future research should monitor the continuation of activities and the wider impacts on primary care.

## 1. Introduction

Antimicrobial resistance (AMR) is a public health threat of global concern. In tackling AMR, a key modifiable factor is the behaviour of healthcare workers and patients towards antimicrobials [1,2]. Antimicrobial stewardship (AMS) interventions and programmes aim to provide resources to improve the capability and motivation for judicious use of antimicrobials [3]. As most antibiotics are prescribed in primary care and dispensed by community pharmacy teams, pharmacists have a key role in AMS that includes checking the safety and appropriateness of prescribed antibiotics and educating patients [4].

Previous qualitative research with community pharmacy staff highlighted AMS roles in providing reassurance and setting patient expectations, in checking antibiotic prescriptions adhere to guidelines, in identifying allergies and interactions, and in discussing prescriptions with prescribers where clinically necessary [5,6]. Reported barriers to AMS included a lack of knowledge and training, time constraints, and lacking access to patients’ medical records [6,7,8].

The pharmacy antimicrobial stewardship intervention (PAMSI) was co-developed with pharmacy staff and other key stakeholders [9,10]. It is based on the capability, opportunity, motivation and behaviour (COM-B) behavioural model [11], and consists of the TARGET Antibiotic Checklist [12], the ‘AMS for Community Pharmacy’ e-Learning module [13], the Antibiotic Guardian pledge [14] and a selection of TARGET information leaflets for patients which include self-care and safety-netting advice for common infections [15]. The TARGET Antibiotic Checklist is designed to be used by pharmacy staff with patients who present with prescription for antibiotics. A section for patients requires them to report their infection, risk factors (such as liver or kidney issues), allergies, and knowledge of antibiotics, which is used by the pharmacy team to tailor their clinical assessments and counselling to patients [9,12]. In this way, the TARGET Antibiotic Checklist provides a framework for AMS actions in the community pharmacy, overcomes barriers around not knowing the patient’s indication for the antibiotic, and provides an opportunity to deliver structured counselling to the patient based on their perceived knowledge gaps [9,10]. PAMSI was evaluated with a sample of staff from a national chain in the UK in 2020. The evaluation found significant improvements in AMS activities reported by community pharmacy staff following the intervention, including checking whether antibiotic prescriptions are appropriate and following local and national guidelines [10].

The English Pharmacy Quality Scheme (PQS) is part of the Community Pharmacy Contractual Framework (CPCF) in England and supports delivery of the National Health Service (NHS) Long Term Plan [16,17]. The overall scheme, which includes several domains, provides financial remuneration to pharmacies who provide NHS services and who choose to participate annually in meeting quality criteria in clinical effectiveness, patient safety and patient experience [18]. Since 2020, the scheme has included a domain for AMS which aims to embed and build on activities over time, including developing an AMS Action Plan, pharmacy staff education, the Antibiotic Guardian pledge, and the implementation of the TARGET Antibiotic Checklist. See Box 1 for the requirements of the AMS domain for the 2021/22 PQS.

Following the positive findings of the small-scale evaluation in 2020, the aim of this study was to assess and describe the national implementation of the TARGET Antibiotic Checklist (which was developed as part of PAMSI) through the PQS incentive scheme, as well as the uptake of AMS e-Learning and the Antibiotic Guardian pledge in 2020/21. The clinical checks completed by the pharmacy team, as well as whether the prescriber was contacted to discuss the prescription, are reported. The accompanying patient data from the TARGET Antibiotic Checklist are reported separately [19]. A follow-up questionnaire with a sample of participating pharmacy staff aimed to explore continued use of the resources and implementation barriers.

Box 1Description of the AMS domain for the 2021/22 PQS [20].
**Infection prevention and control and review of antimicrobial stewardship practice using the TARGET Antibiotic Checklist**
  On the day of the declaration (applies to ALL contractors): Contractors must have reviewed their current practice using the TARGET Antibiotic Checklist, in order to provide tailored advice to patients and promote antibiotic awareness and stewardship.  Using the TARGET Antibiotic Checklist, appropriately trained staff must discuss the antibiotic prescribed with the patient or representative to help ensure safe and effective use. Attempts should be made for this discussion to occur with all patients to promote antimicrobial stewardship.  It may be appropriate to speak to an identified patient representative, family member or member of care staff. If there is a potential risk of antibiotic related adverse effects (for example, change in allergy status) or concerns about the patient’s therapy, the prescriber must be contacted to suggest a review is undertaken and the details of this intervention recorded in the pharmacy PMR.  The pharmacy team should support the patient to reduce the risk of adverse effects arising from ongoing antibiotic therapy and optimise outcomes through education and advice as well as adopting principles of shared decision-making.

## 2. Materials and Methods

### 2.1. Study Design

The service evaluation of the AMS intervention for community pharmacies in England was conducted as part of a national incentive scheme. The PQS requirements were developed by NHS England (NHSE) in collaboration with internal and external stakeholders [20]. The AMS domains relevant for this study were for 2020/21 and 2021/22 of the PQS. In 2020/21, contractors were incentivised to develop a pharmacy level AMS action plan, for staff to complete the ‘AMS for Community Pharmacy’ e-Learning module [13] and to pledge to become an Antibiotic Guardian [14]. The PQS in 2021/22 built on this by incorporating the TARGET Antibiotic Checklist into the AMS action plan [12]. An online questionnaire to collect feedback from a sample of community pharmacy staff two to three months after the 2021/22 PQS asked a range of questions about value of the criteria, barriers to the implementation of the intervention, and continuation of the activities.

### 2.2. Setting and Participants

The setting of this study was community pharmacies in England and the AMS activities that formed part of everyday practice. All community pharmacies providing NHS services in England were eligible to participate in the PQS, providing they met the gateway criteria of the PQS scheme. This included providing an advanced service (the New Medicines Service) which requires them to meet their terms of service and essential service requirements. Exclusion criteria included any non-NHS community pharmacies. Distance selling pharmacies were not excluded. In March 2022 there were 11,232 registered community pharmacies on the NHS England pharmaceutical list. Information about the PQS, including requirements and guidance about the scheme was communicated to contractors through the NHS Business Services Authority via the Drug Tariff and NHSE’s Pharmacy Quality Scheme Guidance.

### 2.3. Data Collection

#### 2.3.1. As Part of the PQS 2020/21

Patient-facing pharmacy staff were required to pledge to be an Antibiotic Guardian and complete the UKHSA ‘AMS for community pharmacy’ e-Learning module, which was hosted on the Health Education England (HEE) platform ‘e-Learning for Healthcare’, between October 2020 and June 2021. HEE collected non identifiable data on e-Learning sessions launches, active learners and the job role of learners.

#### 2.3.2. As Part of the PQS 2021/22

Community pharmacy teams were asked to use the TARGET Antibiotic Checklist for four weeks to reach a minimum sample size of 25 patients or extend its use to eight weeks if the sample size was not met. Any patient presenting at an NHS community pharmacy with an antibiotic prescription was eligible for inclusion. Pharmacy teams could still declare for the domain if they had not reached the sample size after 8 weeks. Contractors needed to complete the domain between September 2021 and May 2022 and submit data from completed checklists on the UKHSA online SnapSurvey (Snap11 professional) platform (Appendix A).

If the community pharmacy had not completed the requirements of the previous 2020/21 PQS, i.e., if they had chosen previously not to participate in PQS or they were a new pharmacy on the pharmaceutical list, they were required to fulfil all the historic PQS as well as the 2021/22 PQS to be eligible for the PQS payment.

The SnapSurvey was a digitalised version of the TARGET Antibiotic Checklist, and pharmacy teams were required to complete a survey for each patient. It also requested the pharmacy ODS code and postcode for verification. The online tool was piloted with 30 pharmacy staff, of whom nine provided further feedback to improve the tool prior to the survey commencing.

#### 2.3.3. A Follow-Up Questionnaire after Completing PQS 2021/22

A follow-up questionnaire hosted on SnapSurvey (Snap11 professional) collected feedback from a sample of staff about their experience of using the TARGET Antibiotic Checklist as part of the 2021/22 PQS (Appendix A). Questions were based on a previously published evaluation and were co-designed and piloted with pharmacists [10]. The survey was open from July to October 2022. Individuals who opted in to receive further information on AMS, or who requested a certificate of completion were invited to complete the questionnaire via email (4628).

### 2.4. Data Analysis

Descriptive analysis of the quantitative TARGET Antibiotic Checklist and follow-up questionnaire data was completed using Microsoft Excel. To describe regional uptake, the postcodes of pharmacies were matched to regions and compared with the list of registered pharmacies from March 2022. For qualitative analysis of the open-ended responses, thematic content coding was completed line by line in QSR NVivo 2020, with each response labelled with codes. The codes were then structured into categories and subcategories, and the frequencies of each occurring theme were reported. Example quotes from the open-ended responses which best illustrated the meaning of the themes were reported.

### 2.5. Ethics

As a service evaluation and as patient identifying data were not collected, institutional ethical approval was not required, as confirmed through the UKHSA Research Ethics and Governance Group and the NHS Health Research Authority decision tool [21]. The data collection tool and follow-up questionnaire were hosted on a secure UKHSA SnapSurvey (Snap11 professional) platform. Informed consent was collected for participating community pharmacy staff via the SnapSurvey data collection tools, and information was provided on the purpose of collecting the data and how they would be used. All files were handled in accordance with the Data Protection Act 2018 and General Data Protection Regulations (GDPR).

## 3. Results

### 3.1. PQS 2020/21 Implementation of AMS E-Learning Module and Antibiotic Guardian Pledge

The AMS e-Learning module was launched on the e-Learning for Healthcare website at the start of 2020 and was freely available for any healthcare staff to complete [13]. The 2020/2021 PQS found that 10,488 pharmacies declared they had completed the AMS e-learning module, comprising of 54,399 registered pharmacists and pharmacy technicians. Data from HEE revealed that there were 148,357 total session launches of the e-Learning module between January 2020 and May 2022. The highest number of sessions were in 2020 (82,493, 56%) and 2021 (54,916, 37%), coinciding with the PQS requirement. There were over 200 different job roles represented among the participants; the pharmacy professions included pharmacist (29,659, 20%), pharmacy technician (12,929, 9%), pharmacy trainee (6391, 4%), pharmacy dispenser (55,760, 38%), and pharmacy assistant (25,129, 17%).

Table 1 shows the number of Antibiotic Guardian pledges from individuals identifying as a member of a pharmacy team located in the UK from 2015 to 2021. Pharmacy teams consisted of academic pharmacists, community pharmacists, pharmacy assistants, pharmacy technicians, primary care pharmacists, and secondary care pharmacists. The introduction of the Antibiotic Guardian pledge as part of the PQS in 2020/21 led to a vast increase in UK pledges from pharmacy teams in 2020, which was sustained in 2021.

### 3.2. PQS 2021/22 Implementation of TARGET Antibiotic Checklist

Of the community pharmacies registered on the NHSE pharmaceutical list in England, 88.5% (9950/11,232) committed to the AMS criteria for the 2021/22 PQS; 74.5% (8374/11,232) submitted data to the online platform by March 2022. All postcode towns, local authorities and regions of England were represented in the data. There were differences in uptake of the PQS activity regionally, as shown in Figure 1. The proportion of participating pharmacies as a percentage of all pharmacies in the different regions of England varied from 64% to 77%.

Data were submitted from 213,105 TARGET Antibiotic Checklists completed with patients collecting antibiotic prescriptions. There was variation in the number of Antibiotic Checklists submitted by the 8374 participating pharmacies (range: 1–181) (Table 2). Of the participating pharmacies, 86% met and/or exceeded the number listed in the criteria (25 TARGET Antibiotic Checklists per pharmacy), 44% exceeded the requirement and 14% submitted fewer than stated in the criteria.

Of 213,105 Antibiotic Checklists submitted, 83% (177,092) were completed in full with all sections completed by pharmacy staff and/or patients. The remaining 17% (36,013) were either partially completed (16%, 35,672), with at least one section missing data or not completed at all (0.2%, 341), with no data for any of the sections. For those partially or not completed, 81% (29,042) provided an explanation for why it was not possible to collect data in full. The main reasons were lack of access to patient information, i.e., a patient representative was collecting on behalf of the patient and therefore did not know the indication and subsequent information (34%, 10,019), or the patient declined to complete the checklist or sections of the checklist (25%, 7263). See Appendix A for full details.

#### 3.2.1. Pharmacy Team Antibiotic Checks

The TARGET Antibiotic Checklist prompted pharmacy teams to conduct and record a series of safety and appropriateness checks against each prescribed antibiotic, and these were completed for 212,124 prescriptions (99.5%). Figure 2 highlights the high frequency of checks reported by the pharmacy teams, particularly the dose, duration, appropriateness of the antibiotic and for patient allergies and interactions (94–95% of prescriptions).

#### 3.2.2. Interventions by Community Pharmacy Teams

Pharmacy teams reported contacting the prescribing clinician to discuss the prescription on 2741 (1.3%) TARGET Antibiotic Checklists. Figure 3 shows a summary of the top five reasons for interventions by pharmacy teams on the TARGET Antibiotic Checklist, and Table 3 breaks this down by the antibiotics dispensed. The majority were for common antimicrobials prescribed in primary care such as amoxicillin and nitrofurantoin. The most common reported reasons that pharmacy teams intervened on a prescription and contacted the prescribing clinician were querying the strength, dose, duration, quantity, and formulation (845, 30%); detection of an allergy (242, 9%); recent antibiotic use (151, 5%); and choice of antibiotics for the indication (144, 5%). No reason was given for 1051 (37%) queries, and in 55 (2%) cases, the antibiotic was not dispensed by the pharmacy team.

Out of 845 queries about strength, dose, duration, quantity, and formulation, 197 were for cases where the prescription did not match the recommended licensed dose, and 101 of these specified the dose was incorrect for the patient age, weight, or gender, mainly where the patient was recorded as a child. There were 139 queries regarding the dose and duration of nitrofurantoin, often used to treat UTIs, and 43/139 (31%) specifically queried the duration not matching current guidance (e.g., ‘Patient is male and over 65, 7-day treatment is required’, ‘Guidance recommends 3 days, but GP wants 7 days’). Long term use and rescue packs were often reported for the prevention of UTI in women who suffer from chronic/recurrent UTIs and patients with COPD, respectively.

There were 242 queries related to allergies, and 146/242 (60%) of these cases specifically stated the antibiotic was changed or not dispensed (clarithromycin was the most common antibiotic to replace a penicillin). The other 96 did not provide information on the outcome of the allergy query, but the majority were for penicillin antibiotics (amoxicillin, flucloxacillin, penicillin V and co-amoxiclav).

Overall, of those who contacted a prescriber, 900 (33%) provided information on the outcome of the discussion. These outcomes were as follows: the prescription dose, formulation, directions, strength or duration was changed for 312 prescriptions (34%); the prescriber confirmed the prescription was appropriate and it did not need to be changed (234, 25%); or an alternative antibiotic was prescribed for 197 prescriptions (21%). Other less reported outcomes included patient information being confirmed (87, 9%), referral of the patient back to the prescriber to reconsult or confirm information (40, 4%), and that the prescription was cancelled (30, 3%).

#### 3.2.3. Pharmacy Staff Follow-Up Questionnaire

There were 105 responses from pharmacy staff for the follow-up questionnaire on the 2021/22 PQS activities (2% response rate of the subset of 4628 of pharmacies invited). Most respondents were a pharmacist manager (40%) or pharmacist (37%), but a range of different types of staff participated. There were similar numbers of respondents who worked in a pharmacy chain and in independent pharmacies. All regions of England were represented. Most respondents (100/105, 95%) stated that they personally contributed to using the TARGET Antibiotic Checklist, and 85%stated they had updated their action plan to include the TARGET Antibiotic Checklist or planned to (7%) since the PQS.

The participant demographics and a full list of qualitative themes can be found in Appendix A, respectively.

The TARGET Antibiotic Checklist was designed so that patients would personally complete the patient section whilst waiting for the prescription to be ready. However, most respondents (75%) reported that staff would ask patients the questions verbally and input answers onto a hard copy and/or directly online. Only 25% reported exclusively asking patients to complete the sections themselves, and 30% used a mix of both approaches depending on the situation. It was reported by 88% of respondents that they aimed to use the TARGET Antibiotic Checklist with any patient with an antibiotic prescription, and 5% aimed to use it with specific patient groups (e.g., patients with acute infections).

There was positive response about the overall value of the TARGET Antibiotic Checklist (Figure 4) for supporting patient care and improving confidence in discussing antibiotic prescriptions with the prescriber. There was a mixed response on whether the time required to use the checklist was feasible (34% agreed, 44% disagreed), and 68% agreed the time was justified by the benefits.

Open-ended responses about the benefits included the staff development achieved by delivering the action plan and training on AMS, and patients benefiting from one-on-one discussions and a better understanding of their antibiotics. A minority found the PQS activities to be less valuable; this was due to the activities already being in place, or a belief that the AMS role should be fulfilled by GPs and other prescribers.


*“The aim is laudable, but this was a PQS hoop-jumping exercise. You’re targeting the wrong people, pharmacies are happy to help patients get best use out of the antibiotics they have been prescribed, but the real levers for change are in the hands of the GPs. We don’t have the time, staff or authority to makes a significant impact”*


Most respondents reported continued use of the resources. Only 11% and 13% reported not using the TARGET Patient Leaflets or TARGET Antibiotic Checklist, respectively, since their declaration of the PQS 2021/22. For the TARGET Antibiotic Checklist, fewer respondents reported that they use it daily, but more have embedded the principles into their daily practice (40%). In regards to the TARGET Patient leaflets, more respondents reported that they use them daily (24%) or sometimes (45%).

While 12 (11%) individuals stated there were no barriers, many discussed the time commitment as a barrier to use of the resources and counseling each patient individually (mentioned 43 times). The respondents discussed a lack of staff and resources. It was stated that provision of printed resources or digital resources which can be used in the pharmacy setting would be favourable.


*“It is really valuable and important but like every part of community pharmacy, every resource is stretched. We are constantly just trying to keep the essentials done”*


Among the respondents, 17% reported issues contacting a prescriber to discuss antibiotic prescriptions; 35% did not need to contact a prescriber during the period of the PQS. Issues faced included the timeliness of contacting the prescriber when a patient was waiting for their medication. Some respondents also reported that the prescriber or staff at the practice did not engage in the discussion, or were dismissive or uninterested.


*“can’t get to speak to them in a reasonable time when patients waiting to take script”*



*“Hard to get a phone back from prescriber and met with stubborn/rude response”*


The patient barriers were that some patients were not willing or were too sick to wait to receive their medication while the clinical checks were being completed, and some were not engaged in the checklist or expected to receive their antibiotics immediately. Another barrier was that some patients were not physically present; either a representative picked up their antibiotics or they were being delivered, and staff did not have the time to follow up on these patients.


*“Although all patients agree that antibiotics should be used appropriately—each individual is not keen to wait and see and prefers to use antibiotics in case they get worse and have to take time out of life to get well”*


## 4. Discussion

### 4.1. Summary

A national incentive scheme led to 8374 pharmacies in England (74% of all eligible pharmacies) actively participating in structured AMS activities whilst dispensing 213,105 antibiotic prescriptions to patients. The participation from every postcode town in England demonstrates the effectiveness of the PQS in achieving widespread national uptake of AMS activities and its respective criteria. The findings suggested high engagement with the scheme as 44% of participating pharmacies submitted more data than was required, and clinical checks by pharmacy teams were completed in high proportions. As of May 2022, there were 51,453 pharmacists and 20,901 pharmacy technicians registered in England across all sectors with the General Pharmaceutical Council (GPhC) [22]; the AMS e-Learning module had been accessed by 42,588 (59%) of all registered pharmacists and pharmacy technicians as of May 2022.

The reasons that prompt pharmacy staff to contact a prescriber to discuss aspects of prescriptions are varied and highlight the clinical role pharmacy teams have in optimising the use of antibiotics. Prescribing information can often be confirmed through the Patient Medication Record (PMR) and/or summary care record (SCR). Pharmacy teams were able to complete appropriate checks to identify possible safety issues, interactions, and treatment failure. Where the pharmacy team reported contacting a prescriber about a given prescription, and where the outcome of the discussion was provided, over half resulted in aspects of the prescription being changed due to involvement of the pharmacy team. This demonstrates the clinical interventions that pharmacy teams can make in antibiotic decision making as well as supporting the AMS efforts of other prescribers.

The antibiotic most often dispensed where pharmacy teams intervened to contact a prescriber was amoxicillin, which is a narrow-spectrum antibiotic used to treat many conditions in primary care. Unsurprisingly, amoxicillin and other penicillin antibiotics were those most queried where an allergy was detected. In a small number of cases, the pharmacy team did not dispense the antibiotic (e.g., where an allergy was detected), but the exact context for this was unknown. Community pharmacy teams also discussed possible unnecessary prescriptions which could be part of a delayed prescribing strategy protocol with prescribers, which is often used, for example, when prescribing antibiotics for sore throats [23].

Some of the queries related to dose and duration were for UTI antibiotics; the guidance for prescribing in women aged 65 years or below is for 3 days, and pharmacy teams intervened, for example, when a patient received a prescription for 7 days [24]. Pharmacy teams also intervened where there was recent antibiotic use; NICE recommends a prescriber to consider switching the antibiotic if the patient has been prescribed the same antibiotic in the last three months [24]. Azithromycin, ciprofloxacin and erythromycin featured in the antibiotics that were discussed with a prescriber, and are the only three that are part of the WATCH group of antibiotics. As expected, no reserve antibiotics featured in this study [25].

The TARGET Antibiotic Checklist asked pharmacy teams to report the reason for querying a prescription, but in many cases the outcome (e.g., if an allergy was confirmed through medical records) and the wider context was not recorded. More in-depth analysis is required to determine the extent of the intervention community pharmacies have made in promoting AMS through their role in discussing prescriptions with prescribers (beyond the scope of this paper).

The follow-up questionnaire findings with a sample of participating community pharmacy staff found that, at least in the short term, most respondents continued to use the resources themselves or the principles introduced, since participating in the PQS. Staff reported improved confidence, development of staff members and a belief that patients benefitted from education. Several barriers to using the resources and participation in AMS in general were reported in the follow-up questionnaire, the main issues being a lack of resources (time and staffing), patient expectations, patients not being present to provide information, and difficulty contacting prescribers to discuss prescriptions.

### 4.2. Comparison with the Literature

AMS in community pharmacy is a growing area of research. Qualitative studies in England, Wales, and Scotland published before 2020 highlighted a lack of awareness of AMS, the role of pharmacy staff and AMR campaigns and interventions [6,7]. In England, this prompted the development of PAMSI which incorporated the pharmacy AMS e-Learning module, the TARGET Antibiotic Checklist and other educational resources for patients. An evaluation of PAMSI in 2019 and 2020 found that community pharmacy staff reported better understanding of AMR and their AMS role, as well as the confidence to provide self-care and adherence advice to patients, and to discuss clinical aspects of prescriptions with prescribers where needed [9,10]. These findings led to the national scale up and implementation of the revised AMS e-Learning module as well as the Antibiotic Guardian campaign through the PQS in 2020/21, which was an important first step to ensure the workforce had the necessary capabilities before introducing the TARGET Antibiotic Checklist the following year.

Qualitative research in France and a UK survey of pharmacy professionals who had pledged to be an Antibiotic Guardian found good knowledge of AMR and motivation to engage in AMS initiatives [26,27,28]. The Antibiotic Guardian survey found significant differences in knowledge of AMR depending on job roles, with pharmacists scoring higher and dispensing staff lower [27]. This suggests some targeted training for different members of teams may be needed to ensure whole team capability for AMS. The PQS requirement was the highest motivation for community pharmacy staff to pledge to the Antibiotic Guardian campaign; however, the researchers found no significant difference in reported AMS behaviour compared with those who had chosen to pledge of their own volition [27]. This suggests that despite the PQS being an incentive scheme, pharmacy staff are engaging fully with the activities, which is supported through our findings that 44% participated at levels above the requirements.

An evaluation of PAMSI in 2020 with 66 community pharmacies found that 84–86% of antibiotic prescriptions were checked by the pharmacy team for allergies, risk factors, dosage, duration and if the antibiotic was appropriate for the indication, and 73% were checked against local prescribing guidelines [10]. This is slightly lower compared with our findings, likely due to the incentivisation associated with the PQS, or because the earlier evaluation took place during winter 2020 when there were high numbers of COVID-19 cases.

A survey in Cornwall found that 18% of pharmacy staff would routinely ask patients their indication as 70% were concerned patients would either not know the clinical name of their infection or would not want to tell the pharmacist [29]. Likewise, a 2016 survey in Northern England found that 30% never ask what infection a patient has, believing it to be sensitive information [30]. The TARGET Antibiotic Checklist overcomes this by allowing patients to write down their infection and/or choose from a list. There is the potential for error as patients may misclassify their infection. Indeed, one of the reasons why staff in our dataset contacted the prescriber was due to the patient’s reported indication not matching the antibiotic prescription. Access to patients’ medical records would overcome these types of queries in an accurate and timely manner, and pharmacy staff have requested this previously [7,28].

Allergies were the second most common reason for the pharmacy team to contact a prescriber in our study. The role of pharmacy staff in checking patient allergies and interactions is well recognised [5,27]. As the TARGET Antibiotic Checklist asks the patient to report an allergy, there is potential for errors; however, the pharmacist can, in most cases, check the summary care record or patient medication record to confirm the allergy. It is estimated that 90–99% of patients with a label of penicillin allergy (PenA) are not allergic when investigated, which can lead to second-line antibiotics being used unnecessarily and increase AMR, healthcare costs and hospital admissions [31]. If there is doubt, it is important that pharmacy teams discuss a potential allergy with prescribers to check if a true allergy was missed during the prescribing stage and so that records can be updated to represent accurate information. Checking allergy status also provides an opportunity to educate the patient about spurious allergies (i.e., nausea and diarrhoea are known side-effects but not allergies).

Effective communication, collaboration and teamwork between pharmacy teams and other healthcare professionals is imperative for AMS. Our follow-up questionnaire findings suggested pharmacy teams face challenges in obtaining prompt access to discuss prescription issues with prescribers whilst under pressure of patients waiting, and a minority reported a lack of ability to directly influence prescribing decisions. A meta-analysis found that antibiotic stewardship programmes that involve pharmacists are effective at improving adherence to antimicrobial prescribing guidelines; therefore, promoting collaboration is key [32]. In some cases, pharmacy staff feel they are not able to question other prescribers’ decisions; for instance, Australian community pharmacy staff reported that they would only try to influence decisions in regards to an allergy or interaction, and this was dependent on their relationship with the prescriber [5]. Other literature highlights the concerns of community pharmacy staff about damaging their relationship with prescribers or a lack of confidence in questioning decisions [6,7,33]. French community pharmacists reported difficult interactions with prescribers, and identified a need to improve pharmacist and GP collaboration and clinical decision tools to support AMS [28]. Interviews with UK community pharmacy staff in 2021 identified concerns that there should be better integration of AMS across the primary care pathway so that the same principles are being discussed with patients consistently [10]. The TARGET Toolkit, which has consistency across the resources for the GP and community pharmacy settings, can support this, but more exploratory research is needed in this area.

Our findings echo the previous evaluation of PAMSI in 2020 which found that pharmacy staff lacked time, resources and staffing to address AMS, and physical resources such as the TARGET Antibiotic Checklist were not feasible for patients who have their prescription delivered or picked up by representatives [10]. Issues of pharmacy staffing, time and resources have been highlighted in other literature [26,28,34]. The feedback survey suggests that most pharmacy teams were reading out the questions to patients rather than having them fill these in directly, as it was designed, which may take more time. Digital and paperless resources are needed to support this process which can be seamlessly built into the prescribing and dispensing journey. The introduction of digital pathways into community pharmacy needs to be understood so that digital interventions and educational web platforms for patients can be introduced.

### 4.3. Strengths and Limitations

A strength of this paper is the large dataset collected through a national scheme, representing large proportion of community pharmacies in England. To date, this is one of the largest datasets investigating AMS activities in community pharmacies. The data give a snapshot of current pharmacy AMS activities in an uncontrolled setting, evaluating the impact of an intervention introduced into daily practice. However, a limitation is that the activities may not be comparable to usual practice as they were introduced through an incentivised scheme. Furthermore, as the data were inputted by pharmacy staff, there are likely to be errors in the data entry process, and it was not feasible to ask participants to double enter data. This required data cleaning, and responses that were ambiguous (e.g., ticked both yes and no options for a question) were removed from the analysis. The data relating to reasons why pharmacy teams contacted a prescriber offer useful insights into interventions by the pharmacy team. An in-depth analysis of these reasons was beyond the scope of this paper and project but could be explored further in future work. The data provided do not include the full context, and in many cases, although the outcome was reported, a clear reason was not given by the pharmacy team when they inputted the data as to why decisions were made. A strength of this paper is the inclusion of follow-up questionnaire data which provide a wider context and understanding of implementation barriers. The responses were from a varied sample representing all regions of England, different pharmacy job roles and types of pharmacies. As the follow-up questionnaire was sent to a sample of participating pharmacies, a limitation is the potential for sampling bias, where it is possible that more enthusiastic individuals participated.

### 4.4. Implications and Conclusions

Through a national incentive scheme, the PQS AMS activities in community pharmacy were promoted effectively. This was demonstrated through high engagement within the scheme, and follow-up questionnaire responses suggest that some AMS activities continued. Future PQS and other activities need to build on this to continue the momentum, and future research should monitor the impact on patient attitudes and behaviours through national public surveys. The PQS for 2022/23 built on the clinical role of pharmacies by embedding shared decision-making tools to support consultations for patients with respiratory and urinary tract infections, before an antibiotic is dispensed [35,36]. All the AMS initiatives have been renewed for the 2023/24 PQS with the addition of advice on safe disposal of unused or expired antibiotics [37]. International audiences could consider adapting these tools to promote AMS. The International Pharmaceutical Federation (FIP) has promoted PAMSI as a case study of good primary health care practice [38].

Our findings further highlight and recognise the essential clinical role that community pharmacy staff have in tackling AMR through AMS interventions and collaborative work with other primary care colleagues, as well as ongoing and continued public education to ensure the safe and appropriate use of antibiotics. These findings add to the evidence that community pharmacy can play a vital role in AMS and support the NHS in England, and internationally, to tackle the significant threat of AMR to public health.

## Figures and Tables

**Figure 1 antibiotics-12-00793-f001:**
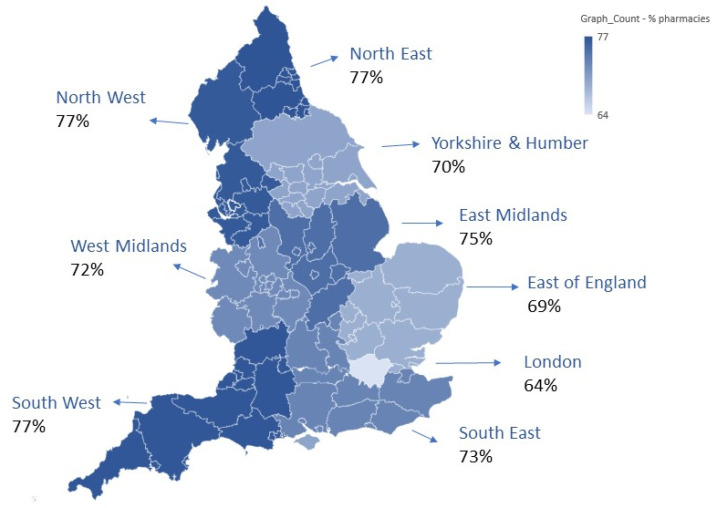
Regional breakdown including proportion of pharmacies who submitted data, as a percentage of all pharmacies in each England region. The colour grade indicates the lowest to highest proportions of pharmacies.

**Figure 2 antibiotics-12-00793-f002:**
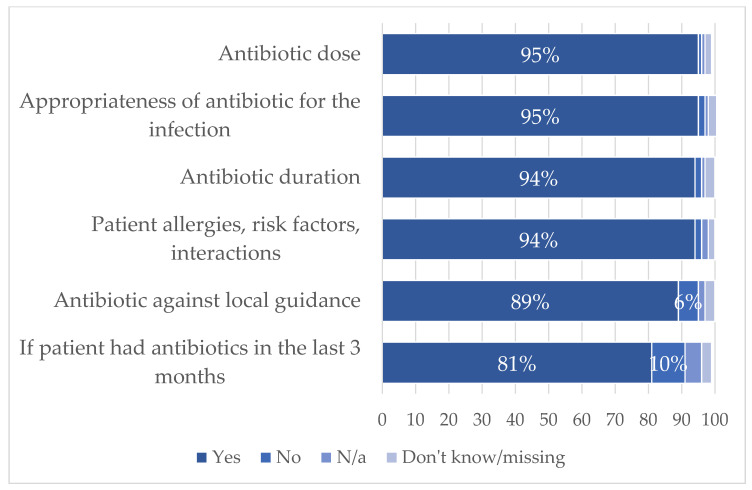
Proportion of antibiotic safety checks completed by the pharmacist or pharmacy team, prompted by the Antibiotic Checklist for 212,124 prescriptions.

**Figure 3 antibiotics-12-00793-f003:**
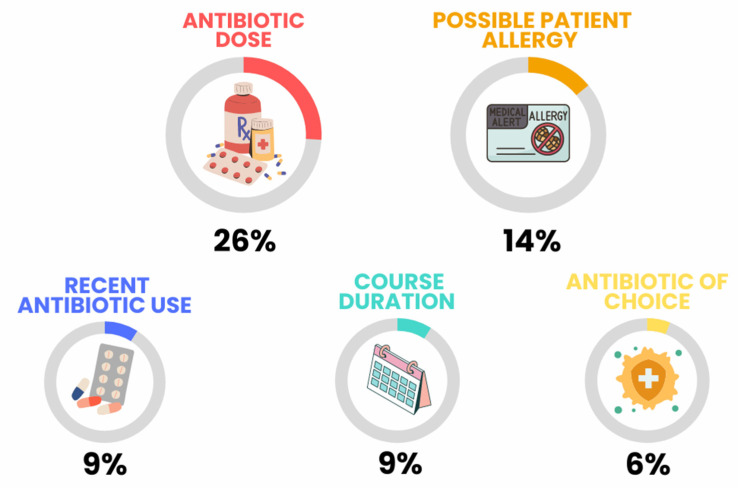
Summary of content analysis themes of why pharmacists contacted a prescriber for 2741 TARGET Antibiotic Checklists.

**Figure 4 antibiotics-12-00793-f004:**
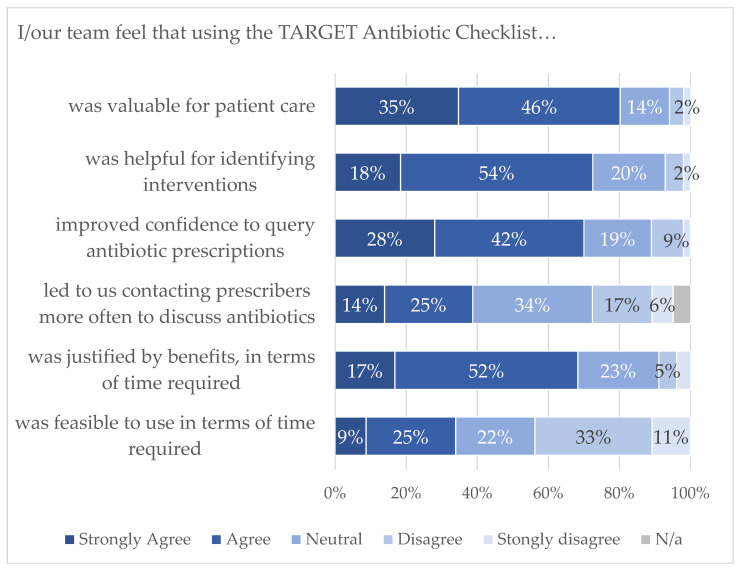
Pharmacy staff feedback on TARGET Antibiotic Checklist.

**Table 1 antibiotics-12-00793-t001:** Number of Antibiotic Guardian pledges from pharmacy teams between 2015 and 2021 from the United Kingdom.

2015	2016	2017	2018	2019	2020	2021	Total
8259	1983	2687	1536	2211	28,369	27,334	72,379

**Table 2 antibiotics-12-00793-t002:** Summary of the numbers of TARGET Antibiotic Checklists submitted by participating pharmacies.

Total number of Antibiotic Checklists Submitted			213,105
Number of Pharmacies Who Submitted Antibiotic Checklists			8374
Range of Antibiotic Checklists Submitted Per Pharmacy			1–181
Completion of Criteria	Number	Percentage of Submitting Pharmacies (n = 8374)	Percentage of Total Pharmacies in England (n = 11,232)
Pharmacies who completed at least 25 Antibiotic Checklists	7225	86%	64%
Pharmacies who completed more than 25 Antibiotic Checklists	3653	44%	32%
Pharmacies who completed fewer than 25 Antibiotic Checklists	1149	14%	10%

**Table 3 antibiotics-12-00793-t003:** Reasons that pharmacy teams intervened on a prescription to contact the prescribing clinician by antibiotic dispensed. The table includes all antibiotics that were recorded on the TARGET Antibiotic Checklist for interventions, including where multiple antibiotics were dispensed, or no antibiotic was dispensed. ^1^ Other reasons included stock availability or omission of information from the prescription. ^2^ ‘Other’ antibiotics were queried less than five times each: levofloxacin, Sofradex, fosfomycin, co-trimoxazole, methenamine hippurate, vancomycin, itraconazole, neomycin, fusidic acid, gentamycin, co-fluampicil, linezolid, rifampicin, cefuroxime, tetracycline, fusidic acid cream, ofloxazin.

Antibiotic	Allergy Detected	Antibiotic Changed or Not Dispensed Due to Allergy	Medicine Interaction	Strength, Dose, Duration, Quantity, Formulation	Multiple Antibiotics Prescribed	Choice of Antibiotic for Indication	Recent Antibiotic Use	Long Term Use, Prophylaxis, Rescue Pack	Possible Unnecessary Antibiotic	Pregnancy, Breast Feeding, Kidney Issue	Patient Factors (Adverse Reaction, Preference)	Safety-Netting/ Referral	Other Reason ^1^	Unknown Reason	Grand Total
Amoxicillin	40	6	7	237	13	31	36	14	7	4	7	5	16	351	774
Nitrofurantoin	4	4	2	139	11	8	41	5	1	22	2	6	5	143	393
Clarithromycin	4	48	29	71	5	8	9	3	1	6	3	-	6	74	267
Flucloxacillin	16	4	3	55	9	15	10	4	2	4	-	6	2	105	235
Doxycycline	3	21	6	60	5	18	18	7	2	5	3	-	2	94	244
Penicillin V	9	6	1	85	6	13	10	-	2	4	-	-	-	63	199
Metronidazole	4	6	1	50	6	7	3	1	-	5	1	-	13	56	153
Trimethoprim	3	3	2	37	8	11	12	3	-	11	2	-	2	44	138
Co-amoxiclav	10	4	1	25	6	8	2	2	1	-	3	-	-	25	87
Cefalexin	-	9	1	8	2	4	2	3	-	4	1	-	-	9	43
Erythromycin	1	8	4	13	2	3	-	1	-	3	1	-	-	7	43
Ciprofloxacin	-	2	2	8	2	8	4	2	-	1	-	1	-	6	36
Clindamycin	-	-	-	12	-	-	-	-	-	-	1	-	-	-	13
Pivmecillinam	-	1	-	8	3	1	2	-	-	1	-	-	2	3	21
Azithromycin	-	-	-	9	-	2	-	2	-	-	-	-	-	1	14
Antibiotic was not dispensed	-	20	-	8	-	-	1	1	6	2	1	-	-	16	55
Unknown antibiotic	2	3	1	4	-	3	-	1	-	1	1	-	2	36	54
Other antibiotic ^2^	^-^	1	3	16	1	4	1	4	-	1	-	-	1	18	50
Grand Total	96	146	63	845	79	144	151	53	22	74	26	18	51	1051	2819

## Data Availability

Data may be available upon reasonable request after contacting the corresponding author.

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
