# Peer review of "The National Implementation of a Community Pharmacy Antimicrobial Stewardship Intervention (PAMSI) through the English Pharmacy Quality Scheme 2020 to 2022"

_antibiotics, 2023, doi:10.3390/antibiotics12040793_

Round 1
Reviewer 1 Report
The present manuscript describes PAMSI through English pharmacies. Even though there are some limitations that the authors have pointed out, the present study concerns a limited number of pharmacists concerning the English health system. This is a key point in whether this study can be of concern to the international community, as in each state there are different rules, especially at this time when UK is no longer a member of the European community.
Author Response
We thank the reviewer for their comments. The points we raise and the tools described within PAMSI have been of interest to the international community, and was recently featured as a case study as part of International Pharmaceutical Federation (FIP) (FIP Primary Health Care World Map – FIP Primary Healthcare), and other countries involved in FIP have expressed interest in the tools. We hope that sharing these findings to an international audience can support other local and international implementation to address the urgent issue of antimicrobial resistance. We have added an additional line to the discussion to note the possible international adaptation of the resources.
Reviewer 2 Report
Dear Authors,
The article is based on a subject of particular importance in the current context of the escalation of microbial resistance to antibiotics, at a rate that causes more and more concern worldwide. And the use of platforms that lead to an extremely important interaction between the patient and the pharmacy staff is an increasingly effective means of personalized patient counseling. This e-learning implementation project with the involvement of all links in the drug chain, from the prescriber to the user, is commendable.
From the point of view of the elaboration of this article, it can be seen that the requirements of the journal are respected in terms of organization by chapters, the summary, the introduction and the chapters are comprehensive and appropriate to the topic addressed.
In section 2.1. the questionnaire practically presents the criteria for inclusion in the study, but the exclusion criteria are not mentioned. For a better picture, the authors should make a brief reference to them.
Line 132, but also later, at lines 401-403 - In the bibliography [15] the article published in 2022 is cited which treats the subject of the present article quite similarly, so the authors should develop and emphasize the progress in the implementation of the program pilot e-learning.
Section 4.3. - the authors present a short swot analysis, focusing on the strengths, but the limitations are not highlighted.
In conclusion, the article deals with an extremely interesting subject that can be an example for other countries to initiate such actions, which can indirectly contribute to a limitation of AMR.
Author Response
We thank the reviewer for their kind comments about the manuscript and we thank them for their suggested changes that we have addressed:
- Inclusion/Exclusion criteria – we have clarified the exclusion criteria for participating community pharmacies within section 2.2 to include non-NHS pharmacies. We have clarified in section 2.3.2 the eligibility criteria for patients (must have prescription for antibiotics).
- In the introduction we have clarified the context of reference 15 being a small-scale service evaluation of PAMSI, which provided the evidence for the national scale implementation described in the present paper. We have also given more detail in the abstract, introduction and in discussion section 4.2, to clarify where this paper sits within the previous published work.
- Thank you, several limitations are discussed in section 4.3 but we have made these clearer in the text.
Reviewer 3 Report
While I do not have any substantive comments on the authors' work, I would recommend revising and improving the references and formatting. For example, I suggest reviewing line 43 or line 64. Additionally, I would recommend improving the clarity of the graphics so that readers can more easily differentiate between the different percentages for each available answer. To achieve this, I recommend using distinct colors (as opposed to shades) while taking care to ensure that the chosen colors are colorblind-friendly.
Author Response
We thank the reviewer for their comments and have made the suggested changes to the referencing and for the bar chart figures, we have kept the shades of blue (this is most accessible for colour blind), however have made the lines more defined, the text larger and the data labels clearer.